# Risk of Malnutrition upon Admission and after Discharge in Acutely Admitted Older Medical Patients: A Prospective Observational Study

**DOI:** 10.3390/nu13082757

**Published:** 2021-08-11

**Authors:** Aino Leegaard Andersen, Rikke Lundsgaard Nielsen, Morten Baltzer Houlind, Juliette Tavenier, Line J. H. Rasmussen, Lillian Mørch Jørgensen, Charlotte Treldal, Anne Marie Beck, Mette Merete Pedersen, Ove Andersen, Janne Petersen

**Affiliations:** 1Department of Clinical Research, Copenhagen University Hospital Amager and Hvidovre, Kettegård Alle 30, 2650 Hvidovre, Denmark; rikke.lundsgaard.nielsen@regionh.dk (R.L.N.); morten.baltzer.houlind@regionh.dk (M.B.H.); juliette.tavenier@regionh.dk (J.T.); line.jee.hartmann.rasmussen@regionh.dk (L.J.H.R.); lillian.moerch.joergensen@regionh.dk (L.M.J.); ctreldal@gmail.com (C.T.); mette.merete.pedersen@regionh.dk (M.M.P.); ove.andersen@regionh.dk (O.A.); janne.petersen.01@regionh.dk (J.P.); 2Department of Clinical Medicine, Faculty of Health and Medical Sciences, University of Copenhagen, Blegdamsvej 3B, 2200 Copenhagen N, Denmark; 3Department of Drug Design and Pharmacology, University of Copenhagen, Universitetsparken 2, 2100 Copenhagen Ø, Denmark; 4The Capital Region Pharmacy, Marielundsvej 25, 2730 Herlev, Denmark; 5Department of Psychology & Neuroscience, Duke University, 2020 W Main St., Suite 201, Durham, NC 27707, USA; 6Emergency Department, Copenhagen University Hospital Amager and Hvidovre, Kettegård Alle 30, 2650 Hvidovre, Denmark; 7Department of Nursing and Nutrition, University College Copenhagen, Sigurdsgade 26, 2200 Copenhagen N, Denmark; anne.marie.beck@regionh.dk; 8Dietetic and Nutritional Research Unit, Herlev-Gentofte University Hospital, Borgmester Ib Juuls Vej 50, 2730 Herlev, Denmark; 9Center of Clinical Research and Prevention and Department of Clinical Pharmacology, Copenhagen University Hospital Bispebjerg and Frederiksberg, Nordre Fasanvej 57, 2000 Frederiksberg, Denmark; 10Section of Biostatistics, Department of Public Health, University of Copenhagen, Øster Farimagsgade 5, 1014 Copenhagen K, Denmark

**Keywords:** malnutrition, risk factors for malnutrition, geriatrics, emergency service, hospital, nutritional assessment

## Abstract

There is a lack of knowledge about malnutrition and risk of malnutrition upon admission and after discharge in older medical patients. This study aimed to describe prevalence, risk factors, and screening tools for malnutrition in older medical patients. In a prospective observational study, malnutrition was evaluated in 128 older medical patients (≥65 years) using the Nutritional Risk Screening 2002 (NRS-2002), the Mini Nutritional Assessment-Short Form (MNA-SF) and the Eating Validation Scheme (EVS). The European Society of Clinical Nutrition (ESPEN) diagnostic criteria from 2015 were applied for diagnosis. Agreement between the screening tools was evaluated by kappa statistics. Risk factors for malnutrition included polypharmacy, dysphagia, depression, low functional capacity, eating-related problems and lowered cognitive function. Malnutrition or risk of malnutrition were prevalent at baseline (59–98%) and follow-up (30–88%). The baseline, follow-up and transitional agreements ranged from slight to moderate. NRS-2002 and MNA-SF yielded the highest agreement (kappa: 0.31 (95% Confidence Interval (CI) 0.18–0.44) to 0.57 (95%CI 0.42–0.72)). Prevalence of risk factors ranged from 17–68%. Applying ESPEN 2015 diagnostic criteria, 15% had malnutrition at baseline and 13% at follow-up. In conclusion, malnutrition, risk of malnutrition and risk factors hereof are prevalent in older medical patients. MNA-SF and NRS-2002 showed the highest agreement at baseline, follow-up, and transitionally.

## 1. Introduction

Worldwide, older patients (≥65 years of age) make up 41–46% of acute admissions, and risk of malnutrition based on nutritional screening is frequent, with a reported prevalence of 35–70% [1,2,3,4,5]. The European Society of Clinical Nutrition (ESPEN) recommended a two-step approach for the diagnosis of malnutrition in 2015, i.e., first, screening for risk of malnutrition with a validated screening tool, and second, assessment for diagnosis of malnutrition [6]. However, the prevalence of malnutrition based on the two-step approach endorsed by ESPEN has not been studied in this population. Nevertheless, malnutrition in older patients is critical as it is related to increased morbidity, readmissions, deteriorating functional ability and quality of life, and mortality [7,8]. In older patients, the etiology of malnutrition is often multifactorial and consists of chronic and acute disease [9] and other commonly known risk factors [10], such as dysphagia [11,12], problems in the oral cavity [12,13] and gastrointestinal tract, polypharmacy [14,15], depression [16], declining cognitive function and declining functional capacity with regards to eating, cooking, and grocery shopping [12].

Acute hospital admissions offer an opportunity to identify malnutrition that would otherwise not be revealed, as 47% of acutely admitted older Danish patients receive only hospital treatment and no other healthcare services [17]. However, the length of stay is usually short [17] and does not leave time for comprehensive nutritional care including treatment of risk factors for malnutrition. Consequently, nutritional care must be provided through collaboration between the hospital and the community services after discharge (i.e., transitionally). 

To design interventional studies with a transitional nutritional intervention, we need information on whether malnutrition or the risk of malnutrition that has been identified during hospitalization still prevails after discharge, or whether it has resolved after treatment of the acute disease. A challenge in this regard is the use of different screening tools during hospitalization and in the community after discharge, as well as the lack of tools that are valid in both settings [18]. Further, we need knowledge on the prevalence of risk factors for malnutrition. Changes in nutritional risk status and prevalence of risk factors for malnutrition from admission until after discharge can provide information on malnutrition screening status and how to intervene. To our knowledge, no studies have comprehensively mapped the risk factors for malnutrition in acutely admitted older medical patients. 

Accordingly, the aim of this study was four-fold: (1) to describe the prevalence of malnutrition and risk of malnutrition upon acute admission and four weeks after discharge; (2) to determine the agreement between malnutrition risk screening tools upon acute admission and four weeks after discharge as well as the transitional agreement within and between malnutrition risk screening tools; (3) to determine the prevalence of malnutrition according to ESPENs 2015 diagnostic criteria; and (4) to determine the prevalence of commonly known risk factors for malnutrition.

## 2. Methods

### 2.1. Study Design and Setting 

The study is part of FAM-CPH, a prospective observational cohort study registered at Clinical Trails.gov (identifier: NCT03052192) with the purpose of investigating medication, malnutrition, chronic inflammation, and biological aging among acutely admitted older medical patients. The participants in the FAM-CPH study were included in the Emergency Department (ED) of Copenhagen University Hospital, Hvidovre, from November 2016 to August 2017. The ED is a 29-bed ward, which is handling acute medical admissions. Minor injuries or traumas are handled in a separate Emergency Room. Other results of the study regarding medication, inflammation and biological aging have been published elsewhere [19,20,21]. Reporting of the study follows the Strengthening the Reporting of Observational Studies in Epidemiology (STROBE) guidelines for cohort studies [22].

### 2.2. Participants and Recruitment

On weekdays between 7 and 8 am, a list of all older medical patients admitted to the ED within the last 24 h and not yet discharged was generated in random order by a computer. Patients on the list were consecutively invited to participate if they fulfilled all the inclusion criteria and none of the exclusion criteria. Inclusion criteria were: age ≥65 years and being acutely admitted. Exclusion criteria were: unable to speak and understand Danish, not Caucasian, unable to cooperate cognitively or physically, in isolation room stay, terminally ill or admitted due to suicide attempt. Invited patients, who accepted participation, provided written informed consent.

### 2.3. Data Collection and Variables

Baseline data were collected by three research assistants (primarily A.L.A. and M.B.H. and occasionally by a research nurse) immediately after inclusion. Four weeks after discharge, home visits were performed to collect follow-up data. Whenever possible, the same research assistant performed both baseline and follow-up data collection.

### 2.4. Baseline Characteristics

Comprehensive baseline characteristics were collected using a structured interview and included information on civil status, level of education, smoking, physical inactivity (less than 2 h of very light physical activity a week) and falls within the previous year. Health-related quality of life was measured by the EQ-5D-5L (with permission from EuroQol Research Foundation). EQ-5D-5L consists of 5 questions rating mobility, self-care, usual activities, pain/discomfort and anxiety/depression on a 5 level scale, and a visual analog scale (VAS-scale from 0–100, with 100 indicating best health) [23,24]. The EQ-5D-5L index was calculated using the EQ-5D-5L Crosswalk Index Value Set [25]. In the Danish Index Value Set, the values range from −0.624 to 1. Functional ability was evaluated by the functional recovery score (FRS), which estimates a person’s degree of dependency in 11 different basic and instrumental activities of daily living [26]. It was chosen for this study, as it was the only device, which to our knowledge evaluates both eating, cooking and grocery shopping. Body weight was measured on a scale (OBH Nordica, model 6256) to the nearest 0.1 kg. If the participant could not stand independently for a sufficient amount of time to measure the weight, a self-reported body weight was registered. Body mass index (BMI, kg/m^2^) was calculated based on body weight and self-reported height. Hand grip strength (kg) was measured as described by Bodilsen et al. [27]. C-reactive protein (CRP) was measured in plasma as part of routine analyses by the Department of Clinical Biochemistry at Copenhagen University Hospital Hvidovre. Diagnosis codes at admission were extracted retrospectively from the medical record. The Charlson Comorbidity Index [28] was calculated without age-correction, and admission diagnoses categorized based on International statistical classification of diseases and related health problems (ICD-10) chapters [29]. 

### 2.5. Screening for Malnutrition and Risk of Malnutrition

Malnutrition and risk of malnutrition were evaluated using the Nutritional Risk Screening 2002 (NRS-2002), the Mini Nutritional Assessment-Short Form (MNA-SF) and the Eating Validation Scheme (EVS). The diagnostic criteria for malnutrition proposed by ESPEN (2015) [30] were applied retrospectively. 

In Denmark, the most commonly applied screening tool during hospitalization is NRS-2002, which is endorsed by ESPEN [9]. NRS-2002 was developed to identify in-patients at risk of malnutrition, who would benefit from a nutritional intervention [30]. NRS-2002 consists of a primary and a secondary screening. The primary screening consists of 4 questions concerning BMI, loss of body weight, decreased dietary intake and critical illness, to which possible answers are yes or no. The secondary screening is performed if there is one yes in the primary screening. It provides a score between 0 and 7 by evaluating the presence or severity of malnutrition (1–3 points) by assessing BMI, recent loss of body weight, recent decline in food intake, and the severity of disease (1–3 points). Further, age ≥ 70 years provides the patient with an additional point. A secondary screening score ≥ 3 indicates that the patient is at risk of malnutrition [30]. NRS-2002 has shown good predictive validity with regard to length of stay [31] and concurrent validity with subjective global assessment (sensitivity: 77% and specificity: 87%) in older hospitalized adults [32].

ESPEN also endorses the use of MNA-SF for screening in older adults [9]. MNA-SF is validated for use both during hospitalization and after discharge and was developed to identify older adults with malnutrition or at risk of developing malnutrition [33]. MNA-SF consists of 6 questions regarding declining food intake, recent weight loss, mobility, acute disease/psychological stress, neuropsychological problems and BMI. It provides a score from 0–14 classifying the person as having a normal nutritional state (score = 12–14), being at risk of malnutrition (score = 8–11) or being malnourished (score = 0–7) [33,34]. Good concurrent validity has been established between MNA-SF and clinical nutritional status (physician-judged) and the full MNA in geriatric inpatients and community dwelling older persons [33]. Concurrent validity has also been established with clinical malnutrition evaluated by a nutritionist in older adults in an acute medical ward [35]. Further, predictive validity has been established with regard to length of stay, in hospital mortality and institutionalization [36,37]. Historically, MNA-SF should be followed by the full-MNA if the MNA-SF score was ≥11, however it was validated as a stand-alone tool in 2009 by Kaiser et al. [34,38]. We applied only the MNA-SF. 

In Danish primary care, the majority (72%) of municipalities use EVS [39]. EVS was developed for use in older persons in primary care and identifies persons who can benefit from a nutritional intervention. The Danish Health Authorities recommend that EVS is applied when an older citizen has had an unintentional weight loss >1 kg or a loss of functional ability [40]. EVS provides a score from 0–2 to classify the person as not at risk of malnutrition (score = 0), at risk of malnutrition (score = 1), or in need of an intervention (score = 2) based on habitual dietary intake, recent weight loss and nutritional risk factors (being dependent on help eating, chewing or swallowing problems, acute disease and acute change in chronic disease) [41]. For the ease of reading and understanding this article, participants with a score ≥ 1 are referred to as at risk of malnutrition.

### 2.6. Diagnostic Criteria for Malnutrition

Retrospectively, we applied diagnostic criteria for malnutrition based on the criteria proposed by ESPEN in 2015 [6], i.e., being at risk of malnutrition according to a validated screening tool (in this study; NRS-2002, MNA-SF, EVS) and having (1) a BMI < 18.5 kg/m^2^ or (2) an unintentional weight loss of more than 10% in an undefined timeframe or an unintentional weight loss of more than 5% within the past 3 months in addition to a BMI < 20 kg/m^2^ if age < 70 years, or a BMI < 22 kg/m^2^ if age ≥ 70 years or a fat free mass index <15 kg/m^2^ in men and <17 kg/m^2^ in women [6]. In this study, we only evaluated weight loss within the past 3 months and therefore used this time frame. Further, we did not measure body composition, which is why we used the BMI criteria and not the fat free mass index. 

### 2.7. Risk Factors for Malnutrition

The following risk factors for malnutrition were assessed: (1) polypharmacy was assessed only at baseline and was evaluated by extracting the prescribed medication from the admission note in the medical record. Polypharmacy was defined as simultaneous use of ≥5 prescribed medications [42]; (2) dysphagia was assessed by the eating assessment gool-10 (EAT-10), which consists of 10 questions rating problems related to the swallowing function. A score of 0–40 points is provided and a score ≥ 3 indicates a need for further examination of presence of dysphagia [43,44]; (3) depression was assessed by The mini geriatric depression score, which consists of 5 questions regarding life satisfaction, boredom, feeling of helplessness, a preference for staying at home, and self-worth. It provides a score between 0 and 5 and a score ≥2 indicates depression and the need for further examination [45]; (4) declining functional capacity regarding eating, cooking and grocery shopping was assessed by a FRS score ≤ 2 in these three parameters; (5) low muscle strength in the lower extremities, defined as less than 5 full stands (assessed by 30 s sit-to-stand test (STS-test), or 5–8 full stands in addition to a habitual gait speed ≤ 0.6 m per second [40], assessed by 4-m gait speed test (GS-test). The STS-test measures the number of times a person manages to stand up fully from a seated position in 30 s without using the arms for support, as described by Jones et al. [46]. If support from the arms was required to stand up, the participant was provided with a score of 0. The GS-test measures the time it takes to walk 4 m at habitual walking pace and was performed as described by Guralnik et al. [47]; (6) problems of the oral cavity and the gastrointestinal tract were assessed by the Eating Symptom Questionnaire (ESQ). The ESQ detects and rates the discomfort (mild to severe discomfort) from 13 symptoms related to the development of malnutrition. These include nausea, vomiting, abdominal pain, diarrhea, constipation, xerostomia, pain which affects appetite, difficulties chewing or swallowing, change in perception of taste and smell, discomfort from smells and pain that hinders eating [48]; (7) the Orientation Memory and Concentration Test [49] (OMC) was applied to evaluate cognitive performance. The OMC consists of 6 questions/tasks to assess orientation in time, recall of a short sentence, backward counting from 20 to zero and the ability to say the months in reverse order. A weighted score from 0–28 is provided, where 28 is worst. An OMC score between 7 and 10 identifies persons with moderately impaired cognitive performance, and a score of 11 or more identifies severe cognitive impairment [49,50,51]. 

### 2.8. Sample Size and Statistics

The calculated sample size for this sub-study was 98 participants, including an expected dropout of 15% of the participants at the follow-up visit. A two-sided non-inferiority test [52] was used to calculate the sample size and was based on the following assumptions: a prevalence of 70% of included patients being at risk of malnutrition [1] at baseline and an expected good agreement between the risk status at baseline and at the 4-week follow-up equivalent to a kappa-value of 0.7. Power was set to 80% and the level of significance to 5% to detect a kappa value statistically smaller or larger than 0.4 (fair agreement). 

Descriptive statistics were used to describe baseline characteristics and risk factors for malnutrition. Continuous variables are presented with a mean and standard deviation (SD) when normally distributed and with median and interquartile range (IQR) when not normally distributed. Differences between continuous variables are tested by *t*-test or Satterthwaite test or a Wilcoxon signed-rank test when appropriate. Categorical variables are presented by percentage, and differences were tested with the Chi-squared test or Fisher’s exact test when appropriate. 

A non-inferiority test was used to confirm the kappa value. A kappa value of 0 represents no agreement between two screening tools, 0.01–0.20 represents none to slight agreement, 0.21–0.40 represents fair agreement, 0.41–0.60 represents moderate agreement, 0.61–0.80 represents substantial agreement, and 0.81–1.00 represents almost perfect agreement [53]. A *p*-value < 0.05 was used as a confirmation that the kappa value was statistically different from a kappa value of 0.4, and the 95% confidence interval (CI) was used to indicate if the kappa value was smaller or larger than 0.4. Agreement between malnutrition risk screening tools is described by the positive predictive value (PPV; i.e., the percentage of participants classified as at risk of malnutrition at baseline who are still classified as such at follow-up), the negative predictive value (NPV; the percentage of participants not classified as at risk of malnutrition at baseline who are still not classified as such at follow-up), sensitivity (percentage of participants at risk of malnutrition who are correctly identified according to the tool of comparison), specificity (percentage of participants not classified as at risk of malnutrition who are correctly identified according to the tool of comparison) and agreement (percentage of participants with or without risk of malnutrition classified in the same way by the screening tool(s)). The PPV was further used to describe the development of the risk factors for malnutrition. The development of the malnutrition screening status within each screening tool was further described by the number of participants who were not at risk of malnutrition at baseline or at follow-up; those who were not at risk of malnutrition at baseline, but developed a risk of malnutrition at follow-up (Acquired); those who were at risk of malnutrition both at baseline and at follow-up (Maintained); and those who were at risk of malnutrition at baseline but not at follow-up (Recovered). 

Data were collected on paper case report forms and double-entered in REDCap (Research Electronic Data Capture, Vanderbilt University, Nashville, TN, USA). Data analysis was performed with SAS Enterprise Guide version 7.1 (SAS Institute Inc., Gary, NC, USA) and R version 3.6.1 (R Foundation for Statistical Computing, Vienna, Austria).

## 3. Results

### 3.1. Participants

A total of 128 participants were included in the study, and 93 completed the 4 weeks follow-up visit, see flowchart, Figure 1. 

### 3.2. Participant Characteristics 

The baseline characteristics of the participants are shown in Table 1. Compared to those who completed the 4 weeks follow-up, participants who were lost to follow-up had a lower functional ability (median FRS, 85 vs. 91, *p* = 0.041), a lower quality of life (median EQ-5D-5L index, 0.615 vs. 0.730, *p* = 0.0031), a higher CRP (median CRP, 14 vs. 52, *p* = 0.013), and higher frequency of low muscle strength in the lower extremities (41% vs. 31%, *p* = 0.0055), see Appendix A. Further, malnutrition or risk of malnutrition and physical inactivity tended to be more prevalent. Otherwise, the two groups were comparable. 

### 3.3. Prevalence of Malnutrition and Risk of Malnutrition and Agreement between Screening Tools

Malnutrition and risk of malnutrition were prevalent at baseline according to MNA-SF, NRS-2002 and EVS (68, 59 and 98%, respectively). At follow-up, malnutrition and risk of malnutrition were still prevalent according to MNA-SF, NRS-2002 and EVS (69, 30 and 85%, respectively). Prevalence of malnutrition according to the ESPEN 2015 diagnostic criteria was 15% at baseline (using MNA-SF or NRS-2002 as the malnutrition screening tool) and 13% at follow-up (using MNA-SF or EVS as the malnutrition screening tool). The prevalences of malnutrition or risk of malnutrition at baseline and the development at follow-up for MNA-SF, NRS-2002 and EVS are shown in Table 2. The full screening tools including answers can be seen in Appendix A. 

The kappa values between the malnutrition screening results at baseline showed slight to moderate agreement. The best overall agreement was found between NRS-2002 and MNA-SF with a classification agreement of 80% (kappa = 0.57 (95%CI: 0.42–0.72), *p* = 0.024 for difference from kappa = 0.4), a sensitivity of 79% and specificity of 82%. NRS-2002 and EVS had a classification agreement of 62% (kappa = 0.07 (95%CI: 0.01–0.15), *p* < 0.001 for difference from kappa = 0.4), a sensitivity of 61% and a specificity of 100%. MNA-SF and EVS had a classification agreement of 70% (kappa = 0.1 (95%CI: −0.01–0.21), *p* < 0.001 for difference from kappa = 0.4), a sensitivity of 69% and a specificity of 100%. 

At follow-up, the kappa values between the malnutrition screening results showed slight to fair agreement between screening tools. The best overall agreement was found between NRS-2002 and MNA-SF, with a classification agreement of 60% (kappa = 0.31 (95%CI: 0.18–0.44) *p* = 0.155 for difference from kappa = 0.4), a sensitivity of 42% and a specificity of 100%. NRS-2002 and EVS had a classification agreement of 45% (kappa = 0.14 (95%CI: 0.06–0.22), *p* < 0.001 for difference from kappa = 0.4), a sensitivity of 35% and a specificity of 100%. MNA-SF and EVS had a classification agreement of 73% (kappa = 0.21 (95%CI: 0.06–0.47), *p* = 0.200 for difference from kappa = 0.4), a sensitivity of 74% and a specificity of 64%.

### 3.4. Transitional Aggrement within and between Screnning Tools 

The transitional kappa value, PPV, NPV, sensitivity, specificity and agreement within and between the three screening tools are shown in Table 3.

The transitional kappa values within and between the screening results indicate slight to moderate agreement. Due to the high dropout-rate, a sensitivity analysis with worst-case scenario was performed with regard to the transitional kappa value. We defined the worst-case scenario as a screening result for malnutrition indicating risk of malnutrition at follow-up for all participants who dropped out. This was deemed a plausible scenario as most participants who dropped out reported illness or lack of energy to participate. The sensitivity analysis (data not shown) with worst-case scenario did not change the *p*-values for the kappa results or the kappa values shown in Table 2. 

### 3.5. Risk Factors for Malnutrition 

The prevalence of the risk factors for malnutrition at baseline and follow-up and their development, described by PPVs, are shown in Table 4. Risk factors for malnutrition were prevalent, with polypharmacy being the most prevalent and the need for help eating being the least prevalent. At baseline, 10 (8%) participants presented with none of the risk factors, 8 (6%) participants presented with 1 risk factor, 16 (13%) participants presented with 2 risk factors, 25 (20%) presented with 3 risk factors, 34 (27%) participants presented with 4 risk factors, 24 (19%) participants presented with 5 risk factors, 8 (6%) presented with 6 risk factors and 2 (1.5%) participants had all of the seven risk factors. The probability that a risk factor for malnutrition still persisted at follow-up, expressed as PPV, ranged between 27–100%.

The probability of having a risk factor for malnutrition when at risk of malnutrition or with malnutrition according to MNA-SF, NRS-2002 or EVS, respectively, is shown in Table 5. Generally, EVS showed a lower probability of having a risk factor than MNA-SF and NRS-2002.

## 4. Discussion

Malnutrition and risk of malnutrition are prevalent among acutely ill older medical patients upon admission and four weeks after discharge, when assessed with the screening tools for malnutrition: MNA-SF, NRS-2002 and EVS. The prevalence ranges between 59–98% at baseline and between 29–88% at follow-up. The prevalence of malnutrition is lower when the diagnostic criteria proposed by ESPEN in 2015 is applied with a prevalence of 15% at baseline and 13% at follow-up. The agreement within and between the applied screening tools for malnutrition at baseline and follow-up was slight to moderate, with NRS-2002 and MNA-SF showing the best agreement at both time points. Moreover, the transitional agreement between the applied screening tools was also slight to moderate, again with MNA-SF and NRS-2002 showing the best agreements. Further, all assessed risk factors for malnutrition were prevalent at baseline and prevailed to a moderate degree at follow-up (PPV ranging from 13–70%). 

### 4.1. Prevalence of Malnutrition and Risk of Malnutrition and Agreement between Screening Tools 

Upon admission, the prevalence of malnutrition and risk of malnutrition according to MNA-SF (68%) is comparable to findings from other studies reporting 48–64% of acutely admitted older medical patients to present with malnutrition or risk of malnutrition upon admission to the ED [3,4,5]. One study, however, found a prevalence of 36% [2]. This lower reported prevalence might be explained by the fact that only 21% of the population in the study by Griffin et al. [2] was scored as suffering from psychological stress or acute disease within the past three months. This prevalence is remarkably low considering the ED setting, where one would expect a higher percentage of patients to suffer from acute disease. We applied MNA-SF 4 weeks after discharge resulting in 91% of the follow-up population scored as suffering from psychological stress or acute disease within the past three months. Application of MNA-SF 13 weeks after discharge would be of interest in future studies to gain knowledge on the prevalence of an MNA-SF score >11 when the question of acute disease within the past three months is no longer applicable to nearly all of the population. We have found no studies applying NRS-2002 to the population of interest in the ED. Two studies in individuals above the age of 18 reported prevalences of 29% and 49% [54,55], which are lower than the prevalence of 59% that we observed. This, we expect to be largely explained by the younger average age. NRS-2002 and MNA-SF presented with the highest agreement at baseline, classifying 80% of the participants similarly with a kappa value of 0.57. These two screening tools are also the only two that are validated in a hospital setting [30,31,32,33,35,36,37] and a high agreement was expected due to similarities on acute disease, BMI and food intake. Velasco et al., however, compared the Full-MNA and NRS-2002 in newly admitted medical and surgical patients above the age of 18 years and found an agreement of 68% and a kappa value of 0.39 [56]. This difference in kappa values between our results and the results of Velasco et al., we expect largely to be due to the higher prevalence of risk of malnutrition according to NRS-2002 in our aged population (59% vs. 35%). Another possible explanation is that we used MNA-SF as a stand alone metric, which may have given a higher prevalence of malnutrition and risk of malnutrition than if the full MNA was applied [3]. However, the full-MNA was deemed too time-consuming considering the ED setting, as it takes approximately 15 min to complete [34]. 

### 4.2. Diagnostic Criteria for Malnutrition

Diagnosis of malnutrition according to the ESPEN 2015 criteria was present in 15% of our study participants at baseline. We are not aware of other studies that have used ESPEN 2015 criteria for malnutrition in acutely admitted older medical patients. A prevalence of malnutrition of 33% and 35% was found by Raupp et al. [54] and Velasco et al. [56] using subjective global assessment (considered a gold standard for assessing presence of malnutrition [57]) despite including acutely admitted patients above the age of 18 years. We would have expected to find at least as many in our population due to advanced age and multimorbidity. However, subjective global assessment [58,59] entails many more aspects that the diagnostic criteria ESPEN proposed in 2015, which could explain the higher prevalence of malnutrition. The lower prevalence of malnutrition in our population may also be due to the fact that we did not collect data on weight loss for more than the past three months. Further, the mean BMI of our population is 26.2, indicating that only very few will fulfill the ESPEN-2015 diagnostic criteria of having a BMI below 20 or 22. Applying a measure of body composition would potentially have led to more participants, including those with a high BMI, being diagnosed with malnutrition. The prevalence of risk of malnutrition and malnutrition as assessed by the screening tools is relatively high compared to the number of participants with a malnutrition diagnosis when assessed by the ESPEN diagnostic criteria (2015). We believe the discrepancy is due to the fact that the ESPEN diagnostic criteria (2015) only encompass phenotypic criteria and no etiologic criteria and therefore does not take dietary intake and severity of disease into account. Further, the lack of convergence in the prevalences may also arise from the difference in aims of ESPEN (2015) and the screening tools, with ESPEN (2015) aiming to diagnose those with malnutrition and the screening tools aiming to identify those at risk of malnutrition, who will not necessarily present with any phenotypic criteria. Applying the Global Leadership Initiative on Malnutrition (GLIM)-criteria for malnutrition [57] would have provided valuable new information and might have resulted in a higher prevalence of malnutrition than the ESPEN (2015) diagnostic criteria for malnutrition, as GLIM assesses more characteristics. Nevertheless, we did not have data on body composition, only the surrogate measure hand grip strength. Additionally, it was not feasible to apply the etiologic criterion for inflammation in our dataset, as we could not retrospectively evaluate if the participants’ inflammation of a mild degree was transient or persisted for a longer period of time.

### 4.3. Transitional Aggrement within and between Screnning Tools and Change in Body Weight 

Different screening tools for malnutrition are used during hospitalization and after discharge in primary care. The transitional agreement of screening tools for risk of malnutrition is important in order to correctly identify patients who would benefit from a transitional nutritional intervention. The best transitional agreement when combining tools was found when using NRS-2002 at baseline and MNA-SF at follow-up, with a kappa of 0.42, a sensitivity of 68% and a specificity of 79%. The agreement of NRS-2002′s results at baseline and follow-up and the agreement of MNA-SF’s results at baseline and follow-up were comparable with this, meaning that these could also be used to identify patients with risk of malnutrition or malnutrition that still prevails after discharge. However, we applied NRS-2002 at follow-up, which is a setting it has not been developed or validated for. Compared to MNA-SF and EVS, the NRS-2002 identifies fewer patients with risk of malnutrition at follow-up as it evaluates disease severity and decreased food intake within the previous week. This and the fact that only 12% of those at risk of malnutrition or with malnutrition according to MNA-SF at baseline had recovered at follow-up (versus 30% when NRS-2002 was applied), implies that MNA-SF used in both settings or the combination of NRS-2002 at baseline and MNA-SF at follow-up are the best tool(s) to evaluate transitional malnutrition or risk of malnutrition. 

When a person is identified with malnutrition or risk of malnutrition in one setting but not in another setting, misunderstandings and miscommunication may arise. In Danish hospitals, NRS-2002 is the most widely used screening tool for risk of malnutrition, whereas EVS is the most widely used tool in primary care. The transitional agreement between NRS-2002 and EVS is reflected by a kappa value of 0.15, a sensitivity of 59% and a specificity of 69%, which indicates that identification of malnutrition and risk of malnutrition is a challenge in the transitional setting. In this regard, it should be noted that re-screening after discharge with EVS is not standard procedure if the patient is discharged with a nutritional plan. However, as many patients in the ED are discharged quickly, they presumably do not receive a nutritional plan and therefore screening after discharge is relevant. In addition, we applied EVS in the hospital setting, which it has not been developed or validated for. This is reflected in the extremely high prevalence of risk of malnutrition on admission according to EVS, and the low transitional kappa values and low specificity when compared to MNA-SF and NRS-2002. The reason for this classification is the fact that a “yes” to the EVS question concerning acute disease will classify the patient as at risk of malnutrition.

### 4.4. Risk Factors for Malnutrtion

The risk factors for malnutrition were prevalent and more than half (66%) of the participants had between three and five risk factors. The risk factors of malnutrition persisted after discharge, with PPVs ranging from 27–84%. These results emphasize the relevance of a multidisciplinary and transitional intervention to prevent or treat malnutrition. Polypharmacy and xerostomia showed the highest prevalences. The high prevalence of polypharmacy is comparable to findings in similar studies reporting prevalences of 64% [5] and 79% [3]. We identified no other studies reporting the prevalence of xerostomia, stressing that this risk factor for malnutrition is overlooked in our patient population. Oral health care also presents as a problem as roughly one in five of our participants at risk of malnutrition reported problems with chewing and pain in the mouth. This is a major issue, as oral health care is not a part of the tax-funded public health care system in Denmark and most other countries [60]. In addition, one in three participants at risk of malnutrition presented with an EAT-10 score of three or above, indicating that they needed further evaluation to identify dysphagia. No other study has, to our knowledge, assessed this prevalence. Further, the prevalence might be higher, as Rofes et al. [43] have shown that a cut-off of two for the EAT-10 score might be more correct. As the only of the applied screening tools, EVS assesses problems of chewing or swallowing and identified 32% of the participants at baseline having such problems. 

Limitations in functional ability regarding cooking and grocery shopping were relevant to one in three participants at risk of malnutrition. This indicates that interventions aiming at improving these specific activities of daily living are relevant in this population. The number of participants who needed help eating was low (*n* = 3). We expect that this is explained by selection bias, as those who declined participation presumably needed more help. Low muscle strength in the lower extremities was prevalent in those who could be assessed. A large part of our population was too ill upon admission for assessment of muscle strength in the lower extremities. Furthermore, the high prevalence of low muscle strength in the lower extremities indicates that the participants at risk of malnutrition or with malnutrition need an intervention aiming to increase muscle strength in the lower extremities. Risk of depression was present in one in four participants at risk of malnutrition, this being a relatively low frequency compared to a prevalence of 84% in a comparable study by Pereira el al [5]. This could be due to cultural differences and differences in the screening tools applied. After medical evaluation of patients at risk of depression, relevant treatment of depression could be initiated during hospitalization and followed-up by the general practitioner. 

Despite exclusion of participants who could not cooperate cognitively, one in four presented with severe cognitive impairment and this persisted in 27% after discharge. Further, roughly half of our population presented with moderate cognitive impairment, which persisted to a high degree at follow-up (PPV of 80%). This must be taken into account when designing a transitional and multifactorial intervention. 

### 4.5. Strengths and Limitations

A major strength of this study is that it is the first of its kind to look at malnutrition prevalence and risk factors in a transitional setting. It provides new and valuable information that emphasizes the importance of nutritional screening and intervention in the ED and after discharge and provides information relevant for designing future interventional studies. The study also has several limitations. A limitation may be the sample size calculation, which assumed a good agreement between the nutritional screening tools. As the agreement was lower than anticipated, the study may lack power. Further, both the nutritional screening tools applied and the questionnaires used to identify causes of malnutrition rely on patient self-reporting. This poses a challenge, as participants may answer, for example, that they do not have a problem with chewing, when in reality they do and have just adapted their diet to the poor chewing ability. This is a limitation of our study as risk factors in this regard may be underreported. Further, the interpretation of the results from EVS upon admission and NRS-2002 in the community setting, is limited due to the lack of validation of the tools in these settings. Using the MNA-SF as a stand-alone tool and not applying the full-MNA may have resulted in higher prevalences of risk of malnutrition and hereby affecting agreement estimates. Another limitation of the study is that the duration of follow-up is shorter (four weeks) than the timespan referred to in the questions regarding weight loss and dietary intake in the NRS-2002 and the MNA-SF (three months) and thus, the two assessments overlap by up to two months. However, one could argue that a given pre-post difference would encompass a change within the follow-up period.

Another aspect that would have strengthened our design is the use of a gold standard for assessing malnutrition. There is no global consensus on how to define a gold standard, but Keller et al. [61] recommend using a comprehensive nutrition assessment as a gold standard and the Full-MNA or SGA as semi golden standards. 

Further, the study population might be subjected to selection bias, as most of the patients we excluded could not cooperate cognitively. Thus, our study does not provide information concerning the most vulnerable patients, which might have altered the prevalences of both nutritional status and risk factors. Lastly, we did not collect data on possible interventions towards malnutrition, which might have been provided in the four weeks from discharge to follow-up. Thus, it is unknown if nutritional support could have affected the screening scores and the nutritional status of the patients and hereby the transitional agreement of the nutritional screening tools. However, as transitional nutritional support is not yet part of standard care, we assume that no intervention has been initiated or impacted the scores during the four weeks following discharge. 

## 5. Conclusions

Malnutrition and risk of malnutrition are highly prevalent among acutely ill older medical patients upon acute admission to the ED and four weeks after discharge. The use of MNA-SF and NRS-2002, alone or in combination, poses the best transitional agreement. However, this agreement was only rated fair to moderate. The high prevalence of risk factors for malnutrition highlights the need for multifactorial interventions to overcome malnutrition in acutely admitted older medical patients. This study provides valuable new knowledge on malnutrition in acutely ill older medical patients which can be used when designing future interventional studies aiming to treat and prevent malnutrition. 

## Figures and Tables

**Figure 1 nutrients-13-02757-f001:**
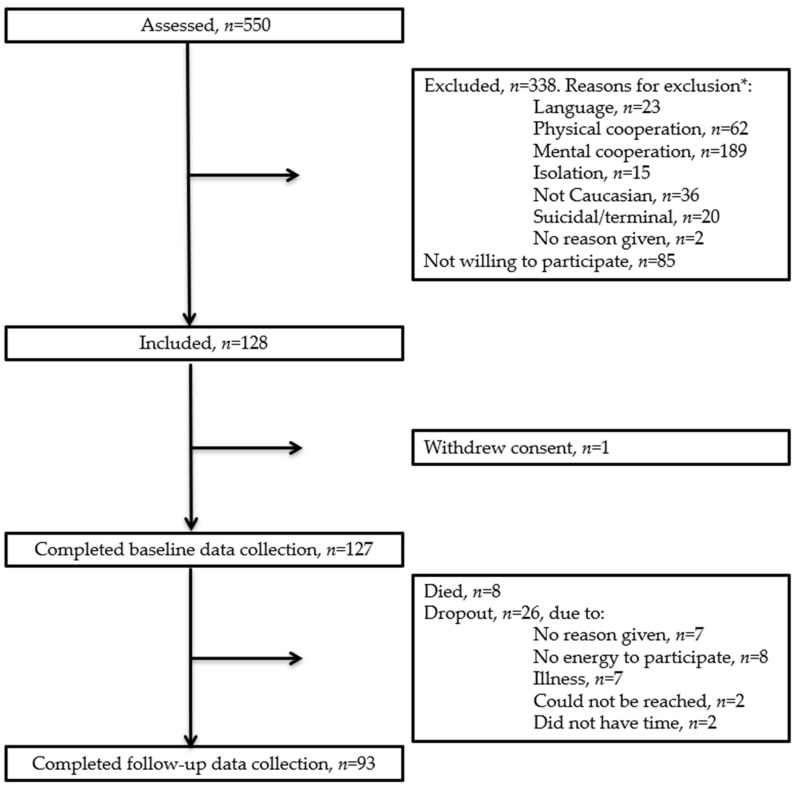
Flowchart. * Patients can have more than one reason for exclusion.

**Table 1 nutrients-13-02757-t001:** Baseline characteristics of the participants.

	All Participants	
Variable	*n*	
Age, years, median (IQR)	127	77.6 (72.3; 85.2)
Sex, male, *n* (%)	127	56 (44)
Smoking, *n* (%)	127	19 (15)
Education		
1–14 years of school, *n* (%)	125	46 (36.8)
Skilled worker, *n* (%)	125	60 (48)
High School +, *n* (%)	125	19 (15.2)
Living alone, *n* (%)	127	84 (66.1)
Body weight (kg), mean (SD)	127	75.9 (19.4)
Height (cm) mean (SD)	127	169.7 (9.9)
Body mass index (kg/m^2^), mean (SD)	127	26.2 (5.5)
Quality of life (EQ-5D-5L index), median (IQR)	125	0.702 (0.493; 0.786)
Charlson comorbidity index, median (IQR)	122	1 (0; 2)
C-Reactive protein (mg/L), mean (SD)	127	22 (5,3; 70)
Maximal hand grip strength (kg), mean (SD)	124	23.2 (10.4)
Functional recovery score, median (IQR)	119	88 (80; 99)
Physically inactive, *n* (%)	126	53 (42.1)
Fallen within previous year, *n* (%)	127	65 (51.2)

Abbreviations: standard deviation (SD), inter quartile range (IQR).

**Table 2 nutrients-13-02757-t002:** Prevalence and development of risk of malnutrition.

	Baseline, *n* (%) *	Follow-Up, *n* (%) *	Development from Baseline to Follow-Up
MNA-SF score ≤ 11	85 (68)	63 (69)	
Maintained a score ≤ 11			49 (57)
Recovered, by gaining a score > 11			10 (12)
MNA-SF score > 11	40 (32)	29 (32)	
Maintained a score > 11			19 (48)
Acquired a score ≤ 11			14 (35)
NRS-2002 score ≥ 3	71 (59)	26 (30)	
Maintained a score ≥ 3			24 (27)
Recovered, by gaining a score < 3			21 (30)
NRS-2002 score < 3	50 (41)	62 (71)	
Maintained a score < 3			37 (74)
Acquired a score ≥ 3			2 (4)
EVS score ≥ 1	122 (98)	78 (85)	
Maintained a score ≥ 1			77 (63)
Recovered, by gaining a score < 1			12 (10)
EVS score < 1	3 (2)	14 (15)	
Maintained a score < 1			2 (67)
Acquired a score ≥ 1			0 (0)

* At baseline 125 completed MNA-SF, 121 completed NRS-2002 and 125 completed the EVS. At follow-up 92 completed MNA-SF, 88 completed NRS-2202 and 92 completed the EVS. Abbreviations: MNA-SF: Mini Nutritional Assessment; NRS-2002: Nutritional Risk Screening-2002; EVS: Eating Validation Scheme.

**Table 3 nutrients-13-02757-t003:** The transitional agreement between and within the nutritional screening tools and tests results for kappa to be different from 0.4.

	MNA-SF at Follow-Up		NRS-2002 at Follow-Up		EVS At Follow-Up	
MNA-SF at Baseline		*p*-Value		*p*-Value		*p*-Value
*n*	92		88		92	
Kappa (95% CI)	0.42 (0.22–0.61)	0.861	0.33 (0.19–0.47)	0.339	0.17 (−0.02–0.36)	<0.001
PPV	83%		44%		90%	
NPV	58%		97%		25%	
Sensitivity	78%		96%		69%	
Specificity	66%		48%		57%	
Agreement	74%		63%		67%	
**NRS-2002 at baseline**						
*n*	88		84		88	
Kappa (95% CI)	0.42 (0.23–0.60)	0.875	0.47 (0.30–0.63)	0.432	0.15 (−0.01–0.30)	<0.001
PPV	87%		53%		92%	
NPV	54%		95%		23%	
Sensitivity	68%		92%		59%	
Specificity	79%		64%		69%	
Agreement	72%		73%		54%	
**EVS at baseline**						
*n*	91		87		91	
Kappa (95% CI)	0.03 (−0.08–0.13)	<0.001	0.02 (−0.01–0.05)	<0.001	0.22 (−0.04–0.48)	0.200
PPV	70%		31%		87%	
NPV	50%		100%		100%	
Sensitivity	98%		100%		100%	
Specificity	4%		3%		14%	
Agreement	69%		32%		87%	

Abbreviations; PPV: positive predictive value; NPV: negative predictive value; CI: confidence interval; MNA-SF: Mini Nutritional Assessment; NRS-2002: Nutritional Risk Screening-2002; EVS: Eating Validation Scheme. Bold indicates the screening tool applied at baseline or follow-up.

**Table 4 nutrients-13-02757-t004:** The prevalence of the risk factors for malnutrition among all participants at baseline and at follow-up and the positive predictive values.

Risk Factor of Malnutrition	Baseline		Follow-Up		PPV, %
	*n*	*n* (%)	*n*	*n* (%)	
Polypharmacy	127	86 (68)			
Risk of dysphagia (EAT10 ≥ 3)	125	34 (27)	92	15 (16)	33
Risk of depression (GDS ≥ 2)	123	27 (22)	91	24 (26)	63
Activities of daily living					
Needs help cooking	125	29 (23)	90	24 (27)	84
Needs help eating	126	1 (0.8)	92	3 (3)	100
Needs help grocery shopping	126	25 (20)	92	21 (23)	82
Low muscle strength in lower extremities					
STS-test < 5, or STS-test > 5 or ≤ 8 + GS ≤ 0.6 m/s	82	43 (52)	76	25 (33)	67
Eating Symptom questionnaire, problems with:					
Chewing	124	21 (17)	93	16 (17)	50
Diarrhea	125	34 (27)	93	19 (20)	46
Xerostomia	125	77 (62)	92	47 (51)	70
Constipation	125	37 (30)	93	24 (26)	63
Nausea	125	50 (40)	93	24 (26)	47
Pain in mouth	124	23 (19)	92	12 (13)	36
Vomiting	125	25 (20)	93	12 (13)	33
Impaired cognition					
Moderately impaired (OMC 7–10) Severely impaired (OMC > 10)	124	68 (55) 28 (23)	90 90	63 (70) 12 (13)	80 27

Abbreviations: EAT-10: Eating Assessment Tool-10, GDS: geriatric depression score, STS: 30 s Sit-To-Stand Test, GS: gait speed, ESQ: Eating symptom Questionnaire, OMC: Orientation Memory and Concentration Test.

**Table 5 nutrients-13-02757-t005:** The probability of having a risk factor for malnutrition at baseline when at risk for malnutrition or with malnutrition according to MNA-SF, NRS-2002 or EVS, respectively.

	At Risk of Malnutrition or with Malnutrition at Baseline According to:		
	MNA-SF Score ≤ 11	NRS-2002 Score ≥ 3	EVS Score ≥ 1
	*n* = 85	*n* = 71	*n* = 122
Risk factor for malnutrition	Probability of having the risk factor, *n*(%)	Probability of having the risk factor, *n*(%)	Probability of having the risk factor, *n*(%)
Polypharmacy	61 (72)	52 (73)	83 (68)
Risk of dysphagia, EAT10 ≥ 3 ^a^	27 (32)	26 (37)	34 (28)
Risk of depression, GDS ≥ 2	24 (29)	18 (26)	27 (23)
Activities of daily living			
Needs help cooking ^a^	26 (31)	22 (31)	29 (24)
Needs help eating ^a^	1 (1)	1 (1)	1 (1)
Needs help grocery shopping	23 (27)	21 (30)	24 (20)
Low muscle strength in lower extremities:			
STS-test < 5, or STS-test >5 or ≤ 8 + GS ≤ 0.6 m/s	34 (69) *	31 (72) **	42 (53) ***
Eating Symptom questionnaire, problems with:			
Chewing ^a^	15 (18)	15 (21)	21 (18)
Diarrhea	28 (33)	25 (35)	34 (28)
Xerostomia	54 (64)	48 (67)	75 (62)
Constipation	26 (31)	26 (37)	36 (30)
Nausea	38 (45)	38 (54)	50 (41)
Pain in mouth^a^	19 (22)	16 (23)	23 (19)
Vomiting	21 (25)	19 (27)	25 (21)
Cognitive Impairment			
Moderately impaired (OMC = 7–10)	41 (49)	35 (50)	67 (56)
Severely impaired (OMC > 10)	25 (30)	21 (30)	28 (23)

* Data are missing for 36 participants, as they could not perform the STS test and/or GS-test at the baseline data collection; ** data are missing for 28 participants, as they could not perform the STS test and/or GS-test at the baseline data collection; *** data are missing for 43 participants, as they could not perform the STS test and/or GS-test at the baseline data collection; ^a^ these risk factors for malnutrition are also a part of the EVS.

## Data Availability

The data presented in this study are available on reasonable request from the corresponding author. The data are not publicly available due to Danish Data Protection laws.

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
