# Peer review of "Risk of Malnutrition upon Admission and after Discharge in Acutely Admitted Older Medical Patients: A Prospective Observational Study"

_nutrients, 2021, doi:10.3390/nu13082757_

Round 1
Reviewer 1 Report
This study is of high quality, evaluating many parameters in a longitudinal cohort, making it possible to assess the adequacy between the different tools on admission and then at 4 weeks of follow-up. Nevertheless I have several comments on the design and on the main objective, as proposed by this study.
1) Logically, a rapid screening test should not include BMI and weight loss; or that the BMI and percentage of weight loss only be calculated only in patients identified as at risk of malnutrition.
2) The duration of follow-up does not make it possible to specifically evaluate the evolution of screening scores such as the NRS and the MNA. In fact, the period of evaluation of the weight loss is, in the 2 cases of 3 months, while the duration of follow-up is 4 weeks.
3) The type of nutritional support during these 4 weeks and its impact on the change in scores are not known, as well as the reasons for non-evaluation at 4 weeks (which represents a large number of patients for such a study)
4) It would be interesting to analyze the reasons for the disparity between the NRS, the MNA and the diagnosis of malnutrition according to the criteria of ESPEN (GLIM). Indeed, the weight loss and BMI criteria are present in the 3 tools. The lack of convergence between these toll and the diagnosis criteria could be explained by the absence of the "etiology" criterion.
5) An important weak point regarding the diagnosis of undernutrition (ESPEN GLIM) is the lack of assessment of muscle mass, for the diagnosis of undernutrition while the major problem encountered in this population is the risk associated with sarcopenia and frailty.
6) This analysis makes it possible to highlight a discordance between the NRS and the MNA in this population, but does not allow, for clinical practice to conclude on the usefulness of one or the other in a screening program to established a diagnosis of undernutrition and its treatment.
7) In the end, we do not understand what this study brings compared to the already existing data on the validity of these screening tool (for example Skipper A, Ferguson M, Thompson K, et al. (2012) Nutrition screening tools: an analysis of the evidence. JPEN J Parenter Enteral Nutr 36: 292-8. Doi: 10.1177 / 0148607111414023.
Reviewer 2 Report
- why did it take you 4 years to publish it? Now the malnutrition criteria recommended by ESPEN are the GLIM, as you comment in the discussion...
- linked to the above, you should note in the abstract, introduction and discussion, that the ESPEN criteria to which you refer are those of 2015.
- were patients with positive screening (or with positive ESPEN criteria) not treated nutritionally? It wouldn't be very ethical... If they were, why isn't it included in the data if it's a very important factor in the evolution of malnutrition that you measure in a second time?
- I believe that the great difference between the prevalence of malnutrition or the risk of it with screening tools, and the GLIM criteria for malnutrition, is not enough justified.
Author Response
Reply to reviewer comments
Reviewer 2
Comment 1:
why did it take you 4 years to publish it? Now the malnutrition criteria recommended by ESPEN are the GLIM, as you comment in the discussion...
Reply 1:
That is a relevant and somewhat painful question. Unfortunately, in fund-based research resources are limited, and things are not always as straight forward as anticipated. Due other obligations of the first author (maternity leave, re-arrangement of resources during the COVID-19 pandemic and running a randomized controlled trial) we have not been able to publish our data until now. However, we do believe that our results are just as relevant now as they would have been when data collection ended and would love to see them published.
We are aware that the criteria recommended by ESPEN are now the GLIM and this merely illustrates that this area is continuously evolving. The GLIM criteria are currently being validated and will be re-evaluated regularly according to the newest literature. We hope that our results can serve as a piece of evidence when different diagnostic criteria are compared, especially as our data on prevalence of malnutrition according to the ESPEN (2015) criteria are the first to be published in our population of interest.
Comment 2:
linked to the above, you should note in the abstract, introduction and discussion, that the ESPEN criteria to which you refer are those of 2015.
Reply 2:
Thank you for making this important point. This has been added all the way through the manuscript.
Comment 3:
Were patients with positive screening (or with positive ESPEN criteria) not treated nutritionally? It wouldn't be very ethical... If they were, why isn't it included in the data if it's a very important factor in the evolution of malnutrition that you measure in a second time?
Reply 3:
Thank you for this relevant comment. The study has been approved ethically as a purely observational study of standard care, meaning that we collected data without summarizing screening scores before all data collection was ended. Further, we applied the ESPEN 2015 diagnostic criteria retrospectively and the criteria therefore did not serve as a screening tool at the time of hospitalization. We agree with the reviewer, however, that not treating patients with positive screening results would not be very ethical. We cannot tell from our data who were treated and who were not, but the fact that some had positive screening results only stresses the importance of screening with the aim of initiating interventions.
We have elaborated more on this subject in the discussion, lines 180-186 (changes are marked in italics): “Lastly, we did not collect data on possible interventions towards malnutrition, which might have been provided in the 4 weeks from discharge to follow-up. Thus, it is unknown if nutritional support could have affected the screening scores and the nutritional status of the patients and hereby the transitional agreement of the nutritional screening tools. However, as transitional nutritional support is not yet part of standard care, we assume that no intervention has been initiated or impacted the scores during the 4 weeks following discharge.”
Comment 4:
I believe that the great difference between the prevalence of malnutrition or the risk of it with screening tools, and the GLIM criteria for malnutrition, is not enough justified.
Reply 4:
Our apologies, but we do not quite understand this comment. As mentioned, we did not have enough data to apply the GLIM criteria for diagnosis of malnutrition (lines 72-73 in the discussion).
However, if we have understood the comment correctly and the reviewer thinks that the great difference between the prevalence of malnutrition or the risk of it compared to the prevalence of malnutrition when using the ESPEN-2015 criteria was not justified enough- we agree with the reviewer.
We have added the following paragraph on this matter in the discussion (lines 52-65) (changes are marked in italics):
“However, Subjective Global Assessment59,60 entails many more aspects that the diagnostic criteria ESPEN proposed in 2015, which could explain the higher prevalence of malnutrition. The lower prevalence of malnutrition in our population may also be due to the fact that we did not collect data on weight loss for more than the past 3 months. Further, the mean BMI of our population is 26.2, indicating that only very few will fulfill the ESPEN-2015 diagnostic criteria of having a BMI below 20 or 22. Applying a measure of body composition would potentially have led to more participants, including those with a high BMI, being diagnosed with malnutrition. The prevalence of risk of malnutrition and malnutrition as assessed by the screening tools is relatively high compared to the number of participants with a malnutrition diagnosis when assessed by the ESPEN diagnostic criteria (2015). We believe the discrepancy is due to the fact that the ESPEN diagnostic criteria (2015) only encompass phenotypic criteria and no etiologic criteria and therefore does not take dietary intake and severity of disease into account.”